# Evaluation and Optimization of Cultural Perception of Coastal Greenway Landscape Based on Structural Equation Model

**DOI:** 10.3390/ijerph20032540

**Published:** 2023-01-31

**Authors:** Yili Yang, Yuxing Chen, Yueyan Liu, Tianyou He, Lingyan Chen

**Affiliations:** College of Landscape Architecture and Art, Fujian Agriculture and Forestry University, Fuzhou 350100, China

**Keywords:** island-type greenway, regional culture, landscape evaluation, cultural perception, structural equation model

## Abstract

The island-type greenway should emphasize the role of maintaining and promoting the island cultural landscape as it serves the function of a general greenway green infrastructure while also having a unique landscape appearance. The northern greenway of Pingtan is used as an example in the paper to illustrate how regional culture is perceived. The first part of the analysis looks at how demographic factors affect the quality of cultural perception. The study reveals that: from a gender perspective, women are more likely than men to perceive regional culture; from an age perspective, people between the ages of 18 and 40 are more likely to perceive regional culture; older people and children are less likely to perceive regional culture; and from a level of education perspective, the higher the education, the stronger the perception. The relationship between tourists’ perceived quality, cognitive image, perceived value, satisfaction, and loyalty to the cultural expression of the greenway landscape is then analyzed by building a structural equation model. According to the findings, visitors’ perceptions of the island’s cultural quality have a positive impact on their cognitive images and perceptions of value, while their satisfaction with the cultural expressions along the coastal greenway has a positive impact on their loyalty.

## 1. Introduction

A greenway is a unique linear green open space with ecological value, transportation value, and historical and cultural preservation value. The cross-connection of greenways to form a staggered and integrated green network is of great significance to improve the relationship between people and nature, safeguard the ecological environment, and improve quality of life [1,2]. Greenways serve a variety of purposes, including those related to the environment, tourism, history, culture, economic value, and so forth. The cultural role of greenways has also drawn more attention from academics in recent years [3]. Regional culture encompasses both material culture and spiritual culture in a broad sense. It is made up of all the material and spiritual accomplishments that a community has made via physical and mental work over the course of long-term historical development, which have been steadily gathered, developed, and sublimated [4]. Regional culture, which also refers to local people’s behavior and mind patterns, is spatially bound. There are distinctions in regional culture because of the various geographical environments, which also affect people’s behavior patterns and ways of thinking [5]. A greenway can be used to convey regional culture, reflecting the distinctive elements of regional culture in the design of the greenway’s landscape, which can gather information about local historical occurrences, folklore, and other aspects, turning the abstract information into concrete design elements and enhancing the excursion [3].

Diverse research on the cultural manifestation of greenways have been conducted by academics. Sharma [6] suggested a method to make the design syntax or materials of greenway design culturally diverse, in order to create landscape variability and unique local characteristics in greenways. Keith [7] proposed to increase the diversity of place landscape and slow walking facilities along greenways to increase social interaction opportunities and cultural benefits. In order to generate collective memory and increase public engagement and vibrancy, some experts also suggested scheduling some cultural festivities and other activities in the public space of greenways or other unused public buildings [8]. Many greenways’ landscape designs place a high value on the expression of details in terms of practical investigation. Utilizing local resources to the fullest extent, slow-moving infrastructure designs such as benches, railings, ground paving, signage, and interpretive signs also incorporate regional cultural traits into the design of landscape nodes to ensure consistency in the cultural expression of greenways [8,9,10,11,12]. The satisfaction level of tourists with the cultural expression of the greenway reflects the quality of the regional culture construction of the greenway, and the satisfaction influence factor reflects the perception and experience of tourists with the greenway. Therefore, the investigation of tourists’ satisfaction and the influencing factors is of positive significance to improve tourists’ visiting experiences, increase regional cultural awareness, and enhance the social influence of the greenway.

Regarding cultural perception, Altunel MC et al. [13] argued that tour quality and satisfaction in cultural tourism affect tourists’ engagement and promotion intention. Liu Tao [14] used a structural equation model to investigate tourists’ perceived quality of red cultural landscape and came to the conclusion that the perceived quality and overall image of red cultural landscape affects tourists’ satisfaction and willingness to revisit. Hultman et al. conducted an empirical study on the interrelationship between destination attributes and tourist satisfaction and tourist destination perceptions [15]; Wu T. E. et al. conducted a tourist satisfaction study on theme, product, and design as the three major factors affecting the attractiveness of salt heritage tourism on the southwest coast of Taiwan [16]; Ramseook-Munhurrun P. et al. used tourists to Mauritius as a research sample to verify the island destinations as influencing factors of tourist satisfaction and loyalty [17]; Chen C. et al. studied the interrelationship between tourist satisfaction and tourism place resources, attractiveness, and competitiveness in Kinmen Island, Taiwan Province, China [18]. While most of the relevant studies focus on parks and scenic areas, this paper explores the interrelationship between tourists’ experiences of cultural expressions of greenway landscape construction and satisfaction and loyalty, with a view to improving the quality of tourists’ experiences and providing reference for the promotion and construction of Pingtan’s regional culture.

There is limited research on the cultural expression of coastal greenways among studies on the cultural functions of various types of greenways. By utilizing their unique natural terrain, islands may fully realize their tourism potential through the building of greenways. Islands are cut off from the mainland; therefore, their cultural traits exhibit both strong ties to the mainland and unique qualities [19]. The zigzag coastline [20], which is a part of the urban open space, has a greenway created along it. The coastal greenway should be built with urban culture, mountain, and sea qualities in mind, and it should make effective use of these features’ spatial characteristics to promote and influence local culture [21]. The creation of the coastal greenway provides a wonderful opportunity to showcase and pass along the intangible and folk cultures of the region. The island’s historical culture can be protected and improved through the mix of tourism and exchanges [22]. In order to make recommendations for the growth of the regional cultural tourism industry, this study uses the northern Pingtan Greenway as the research object. It explores how tourists perceive and are satisfied with the cultural landscape construction of the coastal greenway by building a structural equation model.

## 2. Materials and Methods

### 2.1. Study Area

Pingtan is situated in Fujian Province’s eastern sea, at 119°32′–120°10′ E and 25°15′45″–25°25′45″ N. The entire region is the largest island in the Fujian Province and the fifth largest island in China, with a land area of 371 km^2^ and a sea area of 6064 km^2^. By the end of 2013, the total number of households in the area was 11,831 with a total population of 418,391 [23]. The people of Pingtan migrated from the motherland and have been under the jurisdiction of Fuqing and Fuzhou since the Tang Dynasty. Therefore, the living habits of Pingtan people, such as clothing, food, housing, transportation, and festival customs, are very similar to those of Fuzhou, but some of the ancestors from eastern Fujian, central Fujian, and southern Fujian still retain some of their ancestral customs while integrating into the regional culture of Fuzhou [23]. The unique marine geography and climate environment also created many Pingtan folk customs with strong island flavor [23]. In 2015, Pingtan was awarded the title of Fujian Provincial Geopark. Pingtan Geopark is a comprehensive island geopark with rich typical sea erosion landforms, sea accumulation landforms, and prominent sea erosion relics, supplemented by granite landforms and volcanic landforms, and is a comprehensive island geopark with both marine cultural relics and humanistic landscapes such as stone alley communities. In recent years, the government of Fujian Province has proposed to actively explore cross-strait regional cooperation, establish a regional platform for closer cross-strait cooperation and exchange, set up the Pingtan Comprehensive Pilot Zone, and strive to build Pingtan into a demonstration area for exploring new modes of cross-strait cooperation [23]. As the external gateway of Pingtan Island, the construction of the Pingtan Northern Greenway is conducive to building an internationally competitive tourist destination in Pingtan. Compared with other domestic islands, the policy support for the construction of Pingtan Island is stronger, and its regional culture is simpler and more distinctive, which is more attractive to domestic and foreign tourists.

An essential link in the creation of Pingtan International Tourism Island is the northern Pingtan Greenway, which is situated in Pingtan, Fujian Province. The three national openness and development strategies of “21st Century Maritime Silk Road, Free Trade Zone, and the development of marine economy to establish a strong marine state” may all be implemented in Pingtan thanks to its construction. The project is designed to turn the island of Pingtan into a livable and visitable worldwide tourism destination by utilizing the benefits of nearby tourism resources such island scenery, beaches, bays, and sand views, sea erosion landforms, and Fujian-Taiwan culture. The cultural resources in this region are abundant, and they include archeological sites, wharf culture, the history of the spread of Christianity, the culture of seaside fishing, etc. The cultural and historical values are exceptional. According to the estimation, the total length of the planned greenway in the long term (main line + branch line) is about 37.5 km, the total area of the greenway is 112,369.5 m^2^, the per capita land index is 10 m^2^/person, the instantaneous capacity is 11,236 people, the average passenger turnover rate is 2.2, and the passengers are expected to be about 25,000 per day [23]. However, as development has progressed, it has more served as a means of transportation, and the manifestation of ancient culture is still absent. Relevant studies show that tourists can relax and cultivate themselves by understanding and feeling the regional culture in the process of visiting tourist places, so tourists pay more attention to the construction of regional culture in tourist places. An urgent issue that needs to be resolved in this study is how to better promote area culture and incorporate regional cultural components into the greenway landscape design (Figure 1).

### 2.2. Structural Equation Modeling

#### 2.2.1. Influence Factors

In order to examine how visitors perceive cultural expressions in the landscape design of the northern Pingtan Greenway, this study analyzes red cultural landscapes using Liu Tao’s conceptual paradigm of “perceived quality-perceived value-satisfaction-loyalty” [14]. Tourists from various cultural backgrounds have different perceptions of the landscape [24], and cultural landscape perception is the process of sensory experience and cognition of environmental material elements based on people’s own cultural literacy and way of thinking when they are in a specific cultural atmosphere [25]. Perceived quality, cognitive image, and perceived value are the three key factors this study focuses on. (1) From a landscape gardening perspective, visitor satisfaction is influenced by the visitor’s perception of multiple dimensions, and these factors interact with one another. Additionally, the perceived dimensions have a direct or indirect impact on the gardening components [26], which are the tangible items through which visitors can most readily experience the local culture. The quality of the landscape and the quality of cultural services are determined to be the two main influencing factors of tourists’ perceptions of the site’s quality in this study, along with the site’s unique circumstances. The quality of the landscape primarily comprises two elements: slow walking facilities and cultural perception of landscape points. (2) The perceived worth is primarily demonstrated by the sense of relaxation and enjoyment that visitors to the island’s natural and cultural setting experience. According to Kyoung-Shin et al. [27], visitors’ perceptions of the healing feelings associated with cultural heritage tourism sites have an impact on their satisfaction and propensity to return, and the sites’ vibrant cultural milieu soothes visitors’ bodies and minds. (3) The tourists’ assessment of the island’s historical and cultural development is mirrored in the cognitive image. According to studies, visitors who are open-minded and in awe of the surrounding natural and cultural environment will form an impression of the tourism destination [14]. Numerous tourists are drawn to the island for travel, vacations, relaxation, and stress relief because of its distinctive regional culture and stunning natural surroundings. The experience of traveling is influenced by how people perceive the local culture.

Loyalty affects a visitor’s inclination to return and promote a destination to others, whereas satisfaction is the perception and appraisal of the consequences of the difference between the visitor’s expectations prior to the visit and the experience afterwards [28,29]. Tourists’ propensity to return and recommend visited locations can increase their attractiveness and highlight local traits. The landscape design of the tourist destination and the tourists’ spiritual satisfaction are both considered in this study’s evaluation of satisfaction.

#### 2.2.2. Cultural Experience Satisfaction Model

A multivariate statistical model called a structural equation model may deal with complex interactions between many variables and provide solutions for variables that are difficult to directly observe. The model, which has been extensively employed in the social sciences and other sectors [30], permits both the independent and dependent variables to contain measurement errors in comparison to conventional statistical methods. The goal of this study is to better understand how tourists’ perceptions of the perceived value, perceived quality, and cognitive images of the coastal greenways’ landscape relate to their satisfaction and loyalty. Creating the five sets of hypotheses listed below, then use the structural equation model to verify their validity (Figure 2):

**H1:** 
*Perceived quality is positively correlated with cognitive image.*


**H2:** 
*Perceived quality is positively correlated with perceived value.*


**H3:** 
*Cognitive image is positively correlated with perceived value.*


**H4:** 
*Perceived value is positively correlated with satisfaction.*


**H5:** 
*Satisfaction is positively correlated with loyalty.*


### 2.3. Questionnaire Design

Before distributing the questionnaire, the researchers conducted a field trip to the northern Pingtan Greenway to obtain a preliminary understanding of the design of the landscape on the Greenway and also talked deeply with the tourists to understand their visiting preferences, mainly including 6 aspects: (1) whether you pay attention to the cultural expression of the landscape design on the Greenway; (2) whether you pay attention to the cultural experience of visiting on the Greenway; (3) whether you think the cultural experience of visiting on the Greenway (4) How much do you know about the traditional culture of Pingtan Island and whether the cultural expression of the greenway landscape construction helps you understand the history and culture of Pingtan Island? (5) Compared with the natural scenery of the island, do you think that the inculcation of regional culture helps to relax to a large extent? (6) Compared with other islands, do you think the history and culture of Pingtan Island are simpler and have a greater sense of history? The preliminary research results show that most of the visitors are very concerned about the cultural expression of the landscape construction of the northern Pingtan Greenway and the history and culture of Pingtan Island, and they show great interest in it.

Based on the results of the preliminary research, this questionnaire was designed. Two sections make up the questionnaire study. In the first section, which examines the demographics of tourists, we learn about their fundamental features, such as gender, age, level of education, and country of origin. The second part is the satisfaction evaluation factors of the northern Pingtan Greenway’s cultural expression, which includes 26 measurement items and covers 7 major areas, including the perception of slow walking facilities, the greenway’s landscape, regional cultural services, perceived value, and satisfaction and loyalty. The Five Point Likert Scale is used in the research scale (Table 1).

### 2.4. Questionnaire Survey

The questionnaire survey was conducted in a combination of online and offline methods. The offline survey was selected for June and July 2022, mainly during the peak tourism period of Pingtan Island from 3:00 p.m. to 7:00 p.m. The number of tourists was high during this period, and the new crown epidemic was effectively controlled during this period, so the questionnaires were distributed more efficiently. There were five researchers, all of whom had engaged in social surveys related to greenway planning and were more experienced in interviewing tourists. Both survey methods took the form of questionnaire interviews. Before filling out the questionnaires, information on four aspects of the visitors’ gender, age, education, and place of origin is known, which mainly takes into account that different groups have different comprehensions of regional culture, and the subsequent research results also show this.

Offline survey: five people randomly distributed questionnaires to tourists at various spots on the greenway. Before conducting the survey, the consent of tourists was firstly obtained to understand the degree of tourists’ knowledge about the history and culture of Pingtan Island, to ask tourists’ interest in the history and culture of Pingtan, to focus on this category of people for questionnaire distribution, and to provide necessary explanation and clarification of the survey content to ensure the respondents’ understanding of the purpose of this survey and the questionnaire content. A total of 329 questionnaires were distributed offline, and 308 valid questionnaires were collected after eliminating the invalid questionnaires that were filled out repeatedly. To ensure effective online questionnaire distribution, visitors were also asked about their use of WeChat during the offline research process. The results showed that the interviewees all used WeChat as their main communication software, which facilitated online research.

Online survey: an online survey through questionnaire star was used, and WeChat’s two platforms were used to issue the questionnaires. WeChat is a free application that provides instant messaging services for smart terminals, supporting the rapid delivery of free (with a small amount of network traffic) voice messages, videos, pictures and texts over the Internet across communication carriers and operating system platforms, as well as the use of profiles via shared streaming content. Before issuing the questionnaire, we first confirmed whether the survey respondents have visited the destination and have familiarity with the destination. To confirm this point in the questionnaires, photos and videos were used to help respondents recall the study site overview and explain the meaning of the relevant factors of the questionnaire one by one. A total of 123 copies were distributed online, there were some tourists that could not recall the status of the relevant attractions, so this part of the questionnaire was excluded, and 109 valid questionnaires were collected.

A total of 452 questionnaires were distributed in both ways, and 417 valid questionnaires were returned, for an effective rate of 92.26%. The sample size was in accordance with the required sample size in the structural equation modeling study. SPSS 19.0 and AMOS 24.0 were used for data processing and conceptual model validation analysis. The study followed the basic procedures and principles of scale development in designing the questionnaire. Construct validity was mainly expressed as convergent validity and discriminant validity. The Cronbach’s coefficient of the questionnaire was greater than 0.7, indicating good stability of the measure. The SEM was evaluated based on theoretical perspectives and quality criteria. The maximum likelihood method was used to estimate the parameters of the structural model.

## 3. Results

### 3.1. Demographic Characteristics

The questionnaire survey results in the northern Pingtan Greenway show that (Table 2): the respondents were mainly concentrated in the age range of 18 and 40 years (81.6%), and the education of people with a Bachelor’s degree or above (57.1%) was dominant; there were more visitors who were local residents (58.8%) and relatively few foreign visitors (41.2%); and there were more visitors who were male (50.8%) than female (49.2%).

### 3.2. Effect of Demographic Characteristics on Perceived Cultural Quality

#### 3.2.1. Gender Factor

Using an independent sample *T*-test in SPSS 19.0, the impact of gender on the accuracy of cultural perception was examined. The quality of cultural perception is significantly influenced by gender, as shown in Table 3, and women have a significantly higher level of cultural perception than men.

#### 3.2.2. Age Factor

The results of the one-way ANOVA test for the impact of age on cultural perception quality revealed a statistically significant impact of age on the variability of cultural perception quality (Table 4). Tourists under the age of 18 years and those over the age of 40 years have significantly different cultural perceptions from those groups of travelers, and those between the ages of 18 years and 40 years and those over 40 years have significantly different cultural perceptions as well. The group of travelers between the ages of 18 years and 40 years has the strongest perception of regional culture. The visitors over 40 years had the worst perception, while the visitors under 18 years had the second-best perception.

#### 3.2.3. Education Factor

The findings of the one-way ANOVA test for the influence of different educational backgrounds on the quality of cultural perceptions reveal that there is a substantial difference in how tourists with only a high school diploma and those with a bachelor’s degree or above perceive the local culture. The perspective of local culture is stronger the more education one has (Table 5).

### 3.3. Measurement Model Analysis

#### 3.3.1. Reliability and Validity Analysis

Using SPSS 19.0, the results of the 417 questionnaires that were gathered were examined for consistency and reliability. The findings demonstrated that the questionnaire’s reliability was high, with the overall scale’s Cronbach’s coefficient of 0.989 (>0.700) and the Cronbach’s coefficient being less than 0.989 after removing any question item. The findings of Bartlett’s sphericity test and KMO value analysis on the survey data indicated that the scale’s KMO sampling appropriateness was 0.984 (>0.700). Bartlett’s sphericity test and KMO analysis results revealed that the sample data were appropriate for exploratory factor analysis because the KMO sampling suitability quantity was 0.984 (>0.700), the Bartlett’s sphericity test’s approximate chi-square distribution was 15,977.054, and the significance probability value of P = 0.000 < 0.001 reached a significant level.

The validation factor analysis of the fictitious model using AMOS 24.0 revealed that the standard factor loadings of each item ranged from 0.742 to 0.944, the construct reliability (CR) and average variance extraction (AVE) tests were used to detect the convergent validity of the test variables, all of which were greater than 0.500; the average variance extracted (AVE value) was greater than the standard value of 0.500, indicating that each observed variable could explain the corresponding latent variable more effectively; and the combined reliability (CR value) was greater than 0.800. The fact that all three sets of data satisfy the minimal standard criterion suggests that the questionnaire is very reliable (Table 1).

#### 3.3.2. Structural Equation Model Goodness-of-Fit Analysis

Using the great likelihood method for parameter estimation through AMOS 24.0 software, the structural model was examined for its structural relationships. Table 6 displays the comparison between the calculated results and the standard fitting coefficient values. The measurement model has a good fit degree, as evidenced by the measurement model’s root mean square error of approximation (RMSEA) of 0.069, goodness of fit index (GFI) of 0.862, normative fit index (NFI) of 0.947, and the other indicators all falling within the range of the fitting criteria (Table 7). Further evidence that the model has strong discriminant validity comes from the fact that the root values of each latent variable in the judgment matrix are higher than the correlation coefficients between the latent variables and other latent variables (Table 8).

#### 3.3.3. Path Hypothesis Testing

The results of the path hypothesis test were obtained in this study using the significance level of *p* < 0.050 as the test standard to test and analyze the hypothesis path (Table 9). The outcomes demonstrated that the five path coefficients (*p* values) were all less than 0.05, supporting the validity of the path hypothesis. Figure 3. displayed the final hypothesis model.

## 4. Discussion

### 4.1. Perceived Quality, Perceived Value and Cognitive Image

The standardized path coefficient from perceived quality to cognitive image is 0.98, as seen in the final model diagram (Figure 3). The greenway landscape, the cultural design of the facilities, and the regional cultural services all have a direct impact on how tourists perceive and comprehend the local culture, rising by 0.98 units for every unit increase in perceived quality. While the standardized path coefficient from perceived quality to perceived value is 0.36 and the standardized path coefficient from cognitive image to perceived value is 0.65, respectively, this suggests that the perceived value of regional culture, or the restorative experience of regional culture obtained by tourists, is more difficult to obtain directly through the regional landscape and requires a certain cognitive base of regional culture before the perceived value of regional culture can be attained.

This outcome is mostly driven by the demographics of visitors, many of whom are from outside the area and are not well-versed in the local culture. Visitors also have varying levels of education, age, and other cognitive abilities. These make it difficult for visitors to understand the value of the local culture directly from the greenway landscape, which necessitates accurate interpretation to comprehend the region’s historical changes and to recognize the restoration of physical and mental health during the perception of history. According to Kaplan [43], research findings also demonstrate that the constructed environment has restorative qualities similar to those of the natural environment. Pingtan is very reminiscent of a restorative environment, which influences travelers’ experiences while they are there. This is due to its extensive history and rich culture, as well as the beautiful island scenery.

### 4.2. Perceived Value, Satisfaction and Loyalty

From perceived value to satisfaction, the standardized path coefficient is 0.98. According to research, satisfaction rises by 0.98 units for every unit increase in perceived value, demonstrating a relationship between the pleasure and physical and mental relaxation that may be attained through cultural tourism and satisfaction with the tourist destination. The standardized path coefficient from satisfaction to loyalty is 0.96, meaning that for every unit increase in satisfaction, loyalty increases by 0.96 units, indicating that a person’s level of satisfaction with a particular aspect of the culture of a coastal greenway influences their propensity to return and recommend. The study’s findings concur with those of Wu Jing [44] and Gu Yaqing [45] in that loyalty is largely influenced by satisfaction, which in turn is largely influenced by perceived value. The perceived importance of the island greenway’s cultural expression among visitors also strengthens their bonds with the tourist location, resulting in greater destination loyalty.

## 5. Conclusions

This study investigates how tourists’ cultural perceptions of the northern Pingtan Greenway are now holding up and comes to the following conclusions: (1) Tourists of different genders, ages, and educational levels have very different perspectives on the culture of the coastal greenway; in general, women have a stronger perception of regional culture than men; those between the ages of 18 and 40 have a stronger perception of regional culture; the elderly and children are less likely to feel the regional culture; the higher the education, the wider the experience, the stronger the perception of culture, and the older the tourist, the less likely they are to feel the regional culture. (2) Tourists’ cognitive image and perceived value are favorably impacted by how well they perceive the island culture. On the perceived quality, the three factors barely differ. The architecture of the coastal greenway’s slow-walking facilities, vistas, and cultural service conditions thoroughly reflects the regional culture of the area. Tourists’ perceptions of the local culture are somewhat influenced by the design’s excellence, which both directly and indirectly influences perceived value. (3) Tourists’ assessed the value of the coastal greenway as a cultural manifestation is strongly correlated with contentment. (4) Tourists’ contentment with the coastal greenway’s cultural expression has a favorable impact on their loyalty.

The protection and promotion of the island’s natural and cultural landscape should be highlighted by the coastal greenway, a type of recreational linear open space, on the basis of fulfilling the general greenway’s role as a green infrastructure function. The following recommendations for greenway development are made in light of the study’s examination of tourists’ opinions of cultural manifestations in coastal greenway landscape design.

(1) Make the landscape nodes of coastal greenways more diverse in terms of their cultural themes. Choose to place landscape nodes at the best vantage point for ornamental and distant views so that tourists can appreciate the stunning natural scenery of the island. Appropriately add artificial facilities, the material, color, shape, and other design elements fully integrating into the regional culture, presenting different regional theme characteristics. Enrich the public space of the landscape by including a range of basic service facilities along the greenway, such as food and beverage distribution, cultural exhibitions, entertainment bars, etc. Additionally, introduce social and cultural activities to foster communication between people by increasing the opportunities for social interaction and the benefits of culture. On the other hand, the island’s ecological environment needs to be protected. So that the ecological and cultural functions of the greenway can be more clearly demonstrated, the topography and native vegetation of the site should not be altered during the planning and design of attractions.

(2) Enhance the greenway landscape design’s science education and interpretation framework. The previous study demonstrates that tourists’ perceptions of the cultural expression of the landscape are influenced by age, gender, educational attainment, and other sociocultural factors, and that there are differences in how various groups view regional culture. Regional characteristics can be promoted through voice broadcasting, staff interpretation, video interpretation, and other means. Virtual reality technology can also be used to create digital interactive experiences, virtual scenes, and other means to further tourists’ historical and cultural knowledge, enhancing their understanding of the region’s history and culture without requiring in-depth cultural knowledge.

(3) Make cycling a part of the coastal regional culture traits. Many visitors come to the island for sightseeing because of its stunning natural surroundings. Utilize the creation of the greenway to build a cycling segment with a cultural theme using coastal features, such as sea silk culture, fishing folklore, island village landscape, and other elements. While it is advisable to choose lots with undulating terrain that can easily lead to cycling risks, and to concentrate on the construction of pedestrian and vehicular paths, it is not advisable to place shared bicycles, electric bikes, and other cycling transportation and parking spots to provide convenient cycling conditions. Increase the greenway’s usage rate while lowering risk and meeting a variety of access demands. This enables visitors to take in the local natural beauty and appreciate the local culture, giving them both physical and mental fulfillment.

(4) Reasonable modification of the ratio of commercial, inventive, and spaces for cultural experiences. The coastal greenway’s construction intends to give visitors somewhere to unwind, and a little amount of commercial activity can meet some of their recreational needs. However, it should not go overboard. It should primarily construct private leisure and cultural experience spaces in the greenway experience to feel the island’s natural landscape, experience the island’s historical and cultural changes, and escape the hustle and bustle of urban life.

## Figures and Tables

**Figure 1 ijerph-20-02540-f001:**
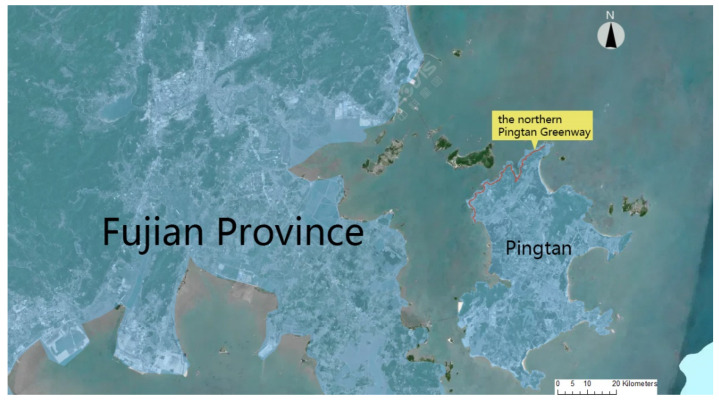
The northern Pingtan Greenway.

**Figure 2 ijerph-20-02540-f002:**
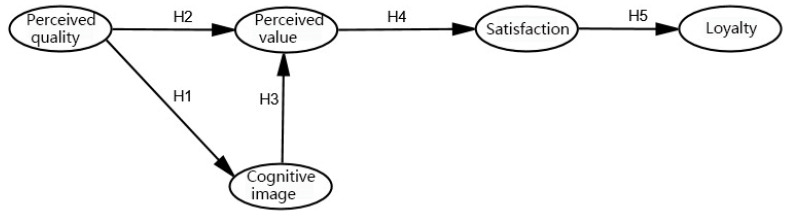
Cultural experience satisfaction model.

**Figure 3 ijerph-20-02540-f003:**
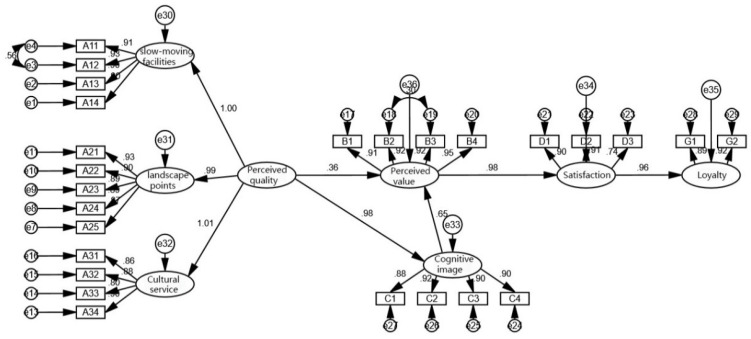
Path hypothesis testing.

**Table 1 ijerph-20-02540-t001:** Measurement scale question setting.

Structure Variables	Source of Observation Indicators
Cultural perception of slow-moving facilities (A1)	Cultural expression of pavement paving [5,8,9,10,11,12,31,32] (A11)
	Cultural expression of signage and interpretation system [5,8,9,10,11,12,31,32] (A12)
	Cultural expression of leisure facilities [5,8,9,10,11,12,31,32] (A13)
	Cultural expression of landscape vignettes [5,8,9,10,11,12,31,32] (A14)
Cultural perception of landscape design (A2)	Distribution of viewpoints [6,9] (A21)
	Regional characteristics of ancient villages [33] (A22)
	Viewpoint geographical appreciation perspective [34] (A23)
	View the regional cultural atmosphere of the site [6,9] (A24)
	Native plant landscape creation [35] (A25)
Cultural service quality (A3)	Greenway Cultural Activities enrichment [8,36,37] (A31)
	Types of transportation to reach various cultural attractions on the greenway [38] (A32)
	Distribution of food and beverage on the greenway [39] (A33)
	Enthusiasm of local residents (A34)
Perceived value (B)	Relieve fatigue from work and study [22,39,40] (B1)
	Increase experience, knowledge [22,39,40] (B2)
	Emotional relief [22,39,40] (B3)
	Rejuvenation [22,39,40] (B4)
Cognitive image (C)	The connotation of traditional culture of the island [14,41,42] (C1)
	The infectious power of traditional culture of the island [14,41,42] (C2)
	The perception of traditional culture of the islands [14,41,42] (C3)
	The specificity of the traditional culture of the island [14,41,42] (C4)
Satisfaction (D)	Very satisfied overall [14,27,42] (D1)
	Very satisfied compared to the ideal island culture [14,27,42] (D2)
	The sense of cultural belonging I get here is what I need [14,27,42] (D3)
Loyalty (G)	Willingness to revisit [27,28,29] (G1)
	Willing to actively recommend this place [27,28,29] (G2)

**Table 2 ijerph-20-02540-t002:** Demographic characteristics of visitors.

	Variables	Frequency	Percentage	Cumulative Percentage
Gender	Male	205	49.20%	49.20%
	Female	212	50.80%	100.00%
Age	<18 years old	13	3.10%	3.10%
	18–40 years old	340	81.60%	84.70%
	Over 40 years old	64	15.30%	100.00%
Academic qualifications	High school and below	91	21.80%	21.80%
	College	88	21.10%	42.90%
	Bachelor’s degree and above	238	57.10%	100.00%
Place of origin	Foreign visitors	172	41.20%	41.20%
	Local residents	245	58.80%	100.00%

**Table 3 ijerph-20-02540-t003:** Independent sample *t*-test for the effect of gender on perceived quality of culture (N = 417).

Dependent Variable	Gender	Average Value	Standard Deviation	Mean Difference	t	*p*
Quality of cultural perception	Female	4.08	0.374	0.11	2.707 *	0.016
	Male	3.97	0.398			

* indicates mean difference significance level of 0.050.

**Table 4 ijerph-20-02540-t004:** One-way ANOVA test for the effect of age on perceived quality of culture (N = 417).

Dependent Variable	I/Year	J/Year	Mean Difference (I-J)	Standard Error	Significance
Quality of cultural perception	<18 years old	18–40 years old	−0.041	0.104	0.577
		Over 40 years old	0.335	0.111 *	0.006
	18–40 years old	<18 years old	0.041	0.104	0.577
		Over 40 years old	0.376	0.055 *	0.000
	Over 40 years old	<18 years old	−0.335	0.111 *	0.006
		18–40 years old	−0.376	0.055 *	0.000

* indicates mean difference significance level of 0.050; I and J refer to age by proxy.

**Table 5 ijerph-20-02540-t005:** One-way ANOVA test for the effect of educational background on the perceived quality of culture (*n* = 417).

Dependent Variable	P	R	Mean Difference (P-R)	Standard Error	Significance
Quality of cultural perception	High School and below	College	−0.098	0.057	0.1170
		Bachelor’s degree and above	−0.200	0.047 *	0.0002
	College	High School and below	0.098	0.057	0.1170
		Bachelor’s degree and above	−0.103	0.048	0.0610
	Bachelor’s degree and above	High School and below	0.200	0.047 *	0.0002
		College	0.103	0.048	0.0610

* indicates mean difference significance level of 0.050; P and R stand in for academic background.

**Table 6 ijerph-20-02540-t006:** KMO and Bartlett’s test.

KMO Sampling Suitability Quantity	Bartlett’s Spherical Test		
	Approximate Cardinality	Degree of Freedom	Significance
0.984	15,977.054	325	0

**Table 7 ijerph-20-02540-t007:** Table of overall fitting coefficients.

Fitting Index	X^2^/df	GFI	AGFI	RMSEA	RMR	NFI	IFI	TLI	CFI	RFI
Adaptation standards	[1,3]	≥0.8	≥0.8	≤0.10	≤0.10	≥0.9	≥0.9	≥0.9	≥0.9	≥0.9
Fitted index value	2.977	0.862	0.832	0.069	0.003	0.947	0.964	0.960	0.964	0.941

**Table 8 ijerph-20-02540-t008:** Measurement scale question fit coefficients.

Structure Variables	Source of Observation Indicators	Standardized Factor Loadings	CR	AVE	Cronbach’s Alpha Value
Cultural perception of slow-moving facilities (A1)	Cultural expression of pavement paving (A11)	0.907	0.951	0.828	0.955
	Cultural expression of signage and interpretation system (A12)	0.933			
	Cultural expression of leisure facilities (A13)	0.902			
	Cultural expression of landscape vignettes (A14)	0.897			
Cultural perception of landscape design (A2)	Distribution of viewpoints (A21)	0.929	0.953	0.801	0.953
	Regional characteristics of ancient villages (A22)	0.897			
	Viewpoint geographical appreciation perspective (A23)	0.889			
	View the regional cultural atmosphere of the site (A24)	0.890			
	Native plant landscape creation (A25)	0.869			
Cultural service quality (A3)	Greenway Cultural Activities enrichment (A31)	0.856	0.919	0.739	0.918
	Types of transportation to reach various cultural attractions on the greenway (A32)	0.880			
	Distribution of food and beverage on the greenway (A33)	0.803			
	Enthusiasm of local residents (A34)	0.896			
Perceived value (B)	Relieve fatigue from work and study (B1)	0.913	0.959	0.854	0.963
	Increase experience, knowledge (B2)	0.919			
	Emotional relief (B3)	0.916			
	Rejuvenation (B4)	0.947			
Cognitive image (C)	The connotation of traditional culture of the island (C1)	0.880	0.945	0.810	0.944
	The infectious power of traditional culture of the island (C2)	0.921			
	The perception of traditional culture of the islands (C3)	0.897			
	The specificity of the traditional culture of the island (C4)	0.902			
Satisfaction (D)	Very satisfied overall (D1)	0.897	0.889	0.730	0.872
	Very satisfied compared to the ideal island culture (D2)	0.913			
	The sense of cultural belonging I get here is what I need (D3)	0.742			
Loyalty (G)	Willingness to revisit (G1)	0.890	0.899	0.817	0.898
	Willing to actively recommend this place (G2)	0.918			

**Table 9 ijerph-20-02540-t009:** Results of path hypothesis testing.

Assumptions	Standardized Path Coefficient	Standard Error	t-Value	*p*-Value	Hypothesis Testing
Perceived quality has a significant positive effect on cognitive image (H1)	0.98	0.035	28.24	***	Established
Perceived quality has a significant positive effect on perceived value (H2)	0.36	0.135	2.621	**	Established
Cognitive image has a significant positive effect on perceived value (H3)	0.65	0.135	4.694	***	Established
Perceived value has a significant positive effect on satisfaction (H4)	0.98	0.035	29.017	***	Established
Satisfaction has a significant positive effect on loyalty (H5)	0.96	0.039	25.558	***	Established

*** indicates *p* is significant at 0.001, ** indicates *p* is significant at 0.01.

## Data Availability

The project has not yet been completed, and the research data are not yet available.

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
