# Peer review of "Evaluation and Optimization of Cultural Perception of Coastal Greenway Landscape Based on Structural Equation Model"

_ijerph, 2023, doi:10.3390/ijerph20032540_

Round 1
Reviewer 1 Report
The work titled "Evaluation and optimization of cultural perception of coastal greenway landscape based on structural equation model" is a research article based on empirical material addressing a case study linking environment, landscape and regional island identity.
The work is in general a reality check of a more general set of hypotheses verified through the specific context of Pingtan island in Fujian Province's eastern sea. The methods used are classical empirical quantitative ones. The article is logically written and arranged and the findings are important for the concrete setting.
The introduction provides sufficient background without being very comprehensive.
Part of the references are more tightly related and used in the framework of the study with influences over the applied concepts and methods. Other part of them is in proximity but not that close to the topicality of the research as an object, themes, development and environmental context.
The research design is appropriate although it needs clear distinction of the contribution of the authors in methodological terms if there is such. I cannot outline such.
The methods are adequately described but there are some important clarification needed for the materials especially around the online and in-person approaches used for the gathering of the survey data.
The results can be better visualized with graphics and more visual and map material of the case - the greenway and its spatial, design and heritage characteristics.
Additional notes can be found in the attached pdf document.

Reviewer 2 Report
The authors study a scientifically really important and relevant topic, and I also appreciate their pragmatic, development-focused approach very much. More studies on this topic would be needed not only in the People’s Republic of China but also globally. However, I think some shortcomings in the manuscript have to be corrected.
First, the authors should be more mindful, I assume, that they want to publish in a global journal. For this reason, from the point of view of the global readership, they should provide a little more information about the island of Pingtan (e.g. with a map on which it would be nice to somehow indicate e.g. Pingtan Greenway).
In connection with the questions of research, for example, it would be good to give the number of permanent residents, the approximate number of tourism and tourists, and their main characteristics in the manuscript, as well (e.g. their number, place of origin, the main attractions and services they prefer (these are also related to my later comments, the question of the representative nature of the questionnaire).
Also, the authors need to address the importance of Pingtan Island and Pingtan Greenway as landslide cultural landscapes for Chinese and local provincial culture in general. Are they in some way considered featured and famous in that sense I mentioned in my previous sentence or not? Because on the one hand, a non-Chinese reader may not know this, and on the other hand, similar development projects presented in the manuscript are implemented globally also, outside the People’s Republic of China in landscapes (islands) of special cultural-historical or even sacral, religious significance.
In addition, there are already similar Greenways and tourism development projects globally that non-Chinese readers may be more familiar with. However, these readers may not know what the significance of Pingtan Island and Pingtan Greenway is within the People’s Republic of China, for them it may be important to be informed about. And, in addition, it would be useful to write it down whether there are similar projects the within the People’s Republic of China, and why the authors chose the given geographical location and case studies for the purposes of the research. More should be written about these aspects, I suppose.
But the main scientific problem I see in the study is that the authors share very little methodological information (questionnaires and interviews) about their research (p. 4). I would definitely suggest a change to this, otherwise, the reader will not know what the results based on research empiricism and the more complex statistics based on it mean exactly and what they measure, according to the author’s intentions, and what conclusions are these empirical results capable of drawing. And more should be written about research limitations in the study as well. Since there is a lot of information to add to this that should be included in the study, I would not list all of them, I just highlight a few of the more important ones.
Questionnaires (both offline and online):
What was the exact type and size of the questionnaire sampling planned (offline and online)?
If the authors planned a representative sample, how representative do the authors consider their sample is for tourists visiting the case study area (which is why more information should be shared about this earlier, I suppose)? Unfortunately, I assume that it is relatively useless for the authors to give their sample in detail, for example, if we do not know to what population and how it should be statistically related.
Who did the offline surveying?
Were there any barriers (e.g. possible effects of COVID)?
How was the online survey technically conducted, e.g. on what platform?
According to the authors, how representative was the online survey for tourists visiting the island?
Interviewing:
How many interviews were conducted, with whom exactly, based on what main topics, and where exactly does the result of the interview appear in the manuscript?
Round 2
Reviewer 2 Report
The manuscript has been improved; however, there are still some points to correct in the manuscript.
Since international readers do not know exactly where it is, why do not you give the global reader a chance to place the scene of your case study in space based on a scale map? Coordinates and textual descriptions without a map are not illustrative enough for this purpose, I am afraid of it.
You have added a few sections to a broader presentation of your case study. However, nowhere in these additional parts do you provide scholarly references and sources, which is scientifically incorrect, as this way the reader will not be able to verify the literature and sources of these passages (e.g. concerning the numeric data mentioned on p. 3.), nor can they investigate further if they are more deeply interested in these additional parts.
“Online survey through questionnaire star, WeChat two platforms to issue questionnaires, before issuing the questionnaire, first confirm whether the survey respondents have visited the destination and familiarity with the destination, to confirm this point, to their fixed-point placement of questionnaires, through photos, videos to help respondents recall the study site overview, and explain the meaning of the relevant factors of the questionnaire one by one, online a total of 123 copies, there are some tourists can not recall the Some visitors could not recall the status of the relevant attractions, so this part of the questionnaire was excluded, and 109 valid questionnaires were collected”:
Please simplify such long sentences (break them into separate sentences). Please correct the edit error I highlighted in bold.
You still provide little methodological information about a part of your research.
I understand that WeChat is popular in the People's Republic of China, but non-Chinese readers probably do not know it that well. For this reason, you should write something about this briefly in general.
And, in addition, wow representative is the 109 valid cases count (which in itself may not be considered too high a number, I assume) you describe of WeChat users who visited the scene of your case study prior to the time point of your research? It should be mentioned.
Do you have any statistics or estimates of how many % of people who visit the case study use WeChat? You should mention it, as well. Because the representativeness of your online sample also depends on the latter, for example.
